# Ultra-thin solid electrolyte interphase evolution and wrinkling processes in molybdenum disulfide-based lithium-ion batteries

Jing Wan[1,2,4], Yang Hao[2,3,4], Yang Shi[1,2], Yue-Xian Song[1,2], Hui-Juan Yan[1,2], Jian Zheng[2,3], Rui Wen[1,2] & Li-Jun Wan[1,2]

Molybdenum disulfide is considered one of the most promising anodes for lithium-ion batteries due to its high specific capacity; however, it suffers from an unstable solid electrolyte interphase. Understanding its structural evolution and reaction mechanism upon charging/discharging is crucial for further improvements in battery performance. Herein, the interfacial processes of solid electrolyte interphase film formation and lithiation/delithiation on ultra-flat monolayer molybdenum disulfide are monitored by in situ atomic force microscopy. The live formation of ultra-thin and dense films can be induced by the use of fluoroethylene carbonate as an additive to effectively protect the anode electrodes. The evolution of the fluoroethylene carbonate-derived solid electrolyte interphase film upon cycling is quantitatively analysed. Furthermore, the formation of wrinkle-structure networks upon lithiation process is distinguished in detailed steps, and accordingly, structure-reactivity correlations are proposed. These quantitative results provide an in-depth understanding of the interfacial mechanism in molybdenum disulfide-based lithium-ion batteries.

[1] Key Laboratory of Molecular Nanostructure and Nanotechnology, Beijing National Laboratory for Molecular Sciences, CAS Research/Education Center for Excellence in Molecular Sciences, Institute of Chemistry, Chinese Academy of Sciences, 100190 Beijing, China. [2] University of Chinese Academy of Sciences, 100049 Beijing, China. [3] Key Laboratory of Organic Solids, Beijing National Laboratory for Molecular Sciences, CAS Research/Education Center for Excellence in Molecular Sciences, Institute of Chemistry, Chinese Academy of Sciences, 100190 Beijing, China. [4]These authors contributed equally: Jing Wan, Yang Hao. Correspondence and requests for materials should be addressed to J.Z. (email: zhengjian@iccas.ac.cn) or to R.W. (email: ruiwen@iccas.ac.cn)

Transition metal dichalcogenides (TMDs, e.g., $MX_2$, where M = transitional-metal element and X = S, Se and Te) are of interest because of their high specific capacity[1] and fundamental properties[2]. Molybdenum disulfide ($MoS_2$) is a common two-dimensional (2D) TMD with a graphene-like lamellar structure in which the interlayer spacing is approximately 0.65 nm. A layered structure with strong covalent bonds within the layers and weak van der Waals forces between layers could provide alkali metal ions ($Li^+$, $Na^+$ and $K^+$) for a transmission channel[3–5]. $MoS_2$ can also be used as a pre-lithiated protective layer for lithium metal anodes in lithium–sulfur (Li–S) batteries[6]. The semiconducting-to-metallic phase transition[7,8] and intercalation chemistry[9] of $MoS_2$ can bring diverse characteristics to $MoS_2$-based battery materials. Thus, $MoS_2$ is considered one of the most promising candidates in lithium-ion batteries (LIBs) owing to its high specific capacity and broad applications. However, low conductivity and severe volume expansion lead to the pulverization of active materials and the accumulation of unstable solid electrolyte interphase (SEI) films during charge–discharge cycles, resulting in decreased reversible capacity, cyclic stability, and rate capability[5,10,11].

SEI films are recognized as the most crucial yet least well-understood phenomena in batteries[12]. Researchers are dedicated to exploring the growth mechanism[13] and composition features[14] of SEI films. Methods of adding additives into electrolytes[15,16], using lithium salts with improved film-formation properties[17], and constructing an artificial SEI film[18,19] have been adopted to establish a stable SEI film. These approaches guarantee the reversible transport of $Li^+$ and prevent the further decomposition of the electrolyte, thus distinctly improving the power performances of batteries[12]. Fluoroethylene carbonate (FEC) has been widely used as an electrolyte additive for graphite[20], silicon (Si)[21,22] and Li metal anodes[23] in advanced Li-ion batteries. It has been further found that adding 10 wt% FEC facilitates the formation of a stable LiF-rich SEI film on $MoS_2$, which can effectively enhance $MoS_2$-based battery performance[10]. Nevertheless, the direct tracking of the nucleation and formation processes of the SEI film on $MoS_2$ to provide deep insights into its interfacial functions and properties has yet to be achieved. Most existing studies on the morphology, chemistry and growth process of SEI films have focused on graphite[24–27], Si[28,29] and Li metal anodes[30–32], but further exploration of the $MoS_2$ electrode is needed to achieve a more detailed and predictive understanding.

Moreover, battery performance is closely related to the kinetics of lithiation/delithiation electrode reactions[11]. Researchers have synthesized $MoS_2$-based nanocomposites[33] to improve battery properties and concentrated on synthesis-microstructure-electrochemical performance relationships[5]. Some real-time and online characterizations have been conducted to reveal the $MoS_2$ electrode reactions. In situ high-resolution transmission electron microscopy (HR-TEM)[34] has demonstrated a phase transition from 2H $MoS_2$ to 1T $Li_xMoS_2$ during lithiation. Analytical tools including operando optics[35], laser confocal microscopy with differential interference microscopy[36] and X-ray absorption spectroscopy[37] have provided additional evidence for the electrochemical behaviour of $MoS_2$. However, the interfacial evolution corresponding to the reaction mechanism during in situ lithiation/delithiation processes is still ambiguous.

Above all, two key fundamental issues at electrode/electrolyte interfaces in $MoS_2$-based LIBs still demand prompt solution to further establish structure–reactivity correlations: first, in situ and quantitative investigation on the initial nucleation and subsequent growth of SEI film and the surface effect of a film-forming electrolyte additive on the electrochemical performance, and second, the nanoscale structural evolution and reaction mechanism of lithiation/delithiation of the $MoS_2$ anode upon charging/discharging. Herein, the interfacial processes on ultra-flat $MoS_2$ anodes in the presence and absence of FEC additives are intensively studied by in situ electrochemical atomic force microscopy (EC-AFM), which closely simulates real batteries and further accurately achieves interfacial properties for a deep understanding of the fundamental mechanism addressed above. To capture the initial nucleation processes of SEI formation and lithiation/delithiation, a large-area ultra-flat monolayer $MoS_2$ electrode prepared by the chemical vapour deposition (CVD) method was employed in the present work. Consequently, the whole nucleation and growth of ultra-thin FEC-derived SEI (the initial thickness is approximately $0.7 \pm 0.1$ nm and subsequently increases to $1.5 \pm 0.7$ nm) formation is in situ and quantitatively elucidated in an FEC-containing system, revealing the effective protection of the electrode from side reactions and volume expansion. The appearance/retention of wrinkles occurring upon lithiation/delithiation testifies to the inherent flexibility of $MoS_2$ and the failure mechanism of $MoS_2$-based LIBs. These results provide not only a fundamental comprehension of the quantitative live formation of ultra-thin SEIs derived from film-forming additives but also direct insights into the structural evolution and reaction mechanism of the $MoS_2$/electrolyte interface upon charging/discharging processes. Therefore, a significant step will advance to the understanding of the dynamic development of ultra-thin and high-quality SEI films and the interfacial engineering and prospective optimization of $MoS_2$-based LIBs.

## Results

**Interfacial processes on ultra-flat monolayer $MoS_2$.** The topological morphology of large-area ultra-flat monolayer $MoS_2$, which is prepared by the CVD method[38], presents a large triangle shape with an average side length of ~25 μm, considering the AFM (inset in Fig. 1a), scanning electron microscopy (SEM) (Fig. 1a) and optical images (Supplementary Fig. 1). The thickness of such a $MoS_2$ electrode is $0.7 \pm 0.2$ nm from the AFM cross-sectional profile along the dashed line indicated in Supplementary Fig. 2a, manifesting the nature and essence of monolayer $MoS_2$. Figure 1b shows a representative cyclic voltammogram (CV) curve of the $MoS_2$ electrode in a 1-butyl-1-methyl-pyrrolidinium bis(fluorosulfonyl)imide ($[BMP]^+[FSI]^-$) ionic liquid (IL) containing $0.5\ mol\ L^{-1}$ lithium bis(fluorosulfonyl)imide (LiFSI) electrolyte with a scan rate of $1\ mV\ s^{-1}$. There is a sharp increase in current upon charging to 1.1 V, and the two reduction peaks at 1.0 and 0.75 V can be attributed to the phase transition from 2H $MoS_2$ to 1T $Li_xMoS_2$ due to $Li^+$ intercalation[34]. The peak at cathodic 0.5 V belongs to the succedent conversion reaction of lithiated $MoS_2$ ($Li_xMoS_2 + (4-x)\ Li^+ + (4-x)\ e^- \rightarrow Mo + 2Li_2S$). Anodic peaks at 1.75 and 2.25 V are ascribed to the decomposition of $Li_2S$ and can be expressed as $Li_2S \rightarrow S + 2Li^+ + 2e^-$[39].

The in situ AFM experiment was first performed on a monolayer $MoS_2$ anode in $[BMP]^+[FSI]^-$ containing 0.5 M LiFSI electrolyte in an electrochemical cell made in-house. Figure 1c shows an AFM image of a monolayer $MoS_2$ electrode at open circuit potential (OCP, ~2.64 V), where an atomically flat terrace can be clearly captured. Subsequently, the structural evolution upon charging at the $MoS_2$/electrolyte interface is further monitored (Fig. 1d–i). Bright nanoparticle (NP) nuclei with a height of $0.8 \pm 0.2$ nm appear at cathodic 1.88 V, as indicated by the yellow arrows (Fig. 1d). The average heights of the NPs gradually increase from $2.2 \pm 0.8$ nm in Fig. 1e to $9.1 \pm 1.6$ nm in Fig. 1f, manifesting their growth and accumulation at 1.39 V on both basal planes and edge sites. This process could be ascribed to the initial nucleation and further development of SEI film, considered as an NP-shaped SEI, consisting of reduction products

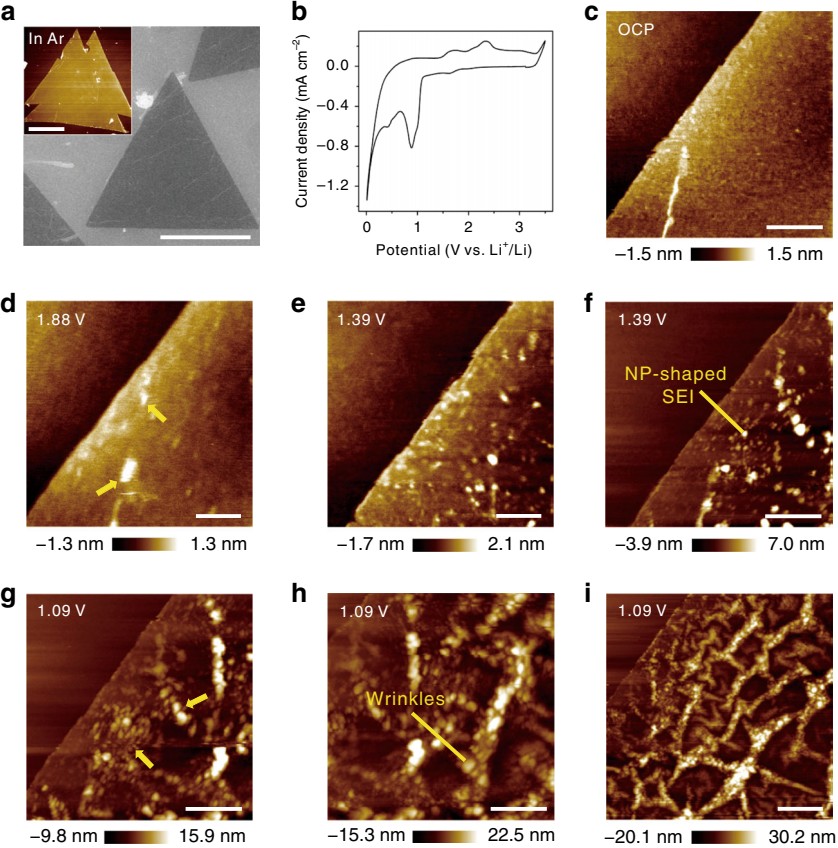

**Fig. 1** Structural evolution at the ultra-flat monolayer MoS$_2$/electrolyte interface via in situ AFM. **a** AFM (inset view) and SEM images of the topological morphology of the large-area ultra-flat MoS$_2$ electrode. **b** Cyclic voltammogram curve of the MoS$_2$ electrode in [BMP]$^+$[FSI]$^-$ containing 0.5 M LiFSI electrolyte at a scan rate of 1 mV s$^{-1}$. AFM images on the MoS$_2$ electrode at different potentials of **c** OCP, cathodic **d** 1.88 V, **e**, **f** 1.39 V and **g–i** 1.09 V. The scale bars are 10 μm in **a**, 200 nm in **d, e**, 500 nm in **c**, **f–h** and 1 μm in **i**

of [FSI]$^-$ in the electrolyte[17]. It is apparent that the NP-shaped SEI film is loosely distributed on the MoS$_2$ surface with the interphasial morphology of dispersed NPs. Nonetheless, the SEI film originating from LiFSI-based electrolyte is still relatively uniform and of high quality compared with the heterogeneous nature and uneven distribution of films in industry-standard electrolytes[26], elucidating the superior film-forming properties of LiFSI salt. When the potential approaches 1.09 V, bright NPs accumulate continuously on the platform, and bulges (marked by yellow arrows) appear uniformly at the electrode surface along certain directions (Fig. 1g). As lithiation proceeds, the bulges grow, propagate and branch as wrinkles (Fig. 1h), and ultimately evolve into planar nanofold-structure networks at the interface (Fig. 1i). Detailed ravines and valleys of mesh-distributed nanofolds are distinctly distinguished as surface features developed from wrinkles in the three-dimensional (3D) AFM images, as shown in Supplementary Fig. 3. During the delithiation process, the wrinkling maintains a similar morphology, and the volume expansion cannot shrink reversibly after delithiation, as shown in Supplementary Fig. 4.

Regarding the driving force of the wrinkling, the in-plane compressive stresses of MoS$_2$ due to ion implantation[40,41] have been proposed. It was revealed that a phase transition process takes place from a semiconducting 2H-MoS$_2$ to metallic 1T-Li$_x$MoS$_2$ upon lithiation[2,42–45]. At the early lithiation stage, Li$^+$ embeds into the tetrahedral coordination centre of the S–S, causing charge transfer from intercalated Li$^+$. Due to the co-intercalation of ions and electrons, the coordination structure of metal Mo is converted from the trigonal prism to octahedron, and

the corresponding space group is transformed from $P6_3/mmc$ to $P$-3$m$1 (ref. [46]), thus producing a 2H–1T phase transition to reduce the energy of the system[47]. Associated first-principles calculations elucidate that the essence of the phase transition is the interaction of intrinsic doping and electron-phonon coupling[48]. HR-TEM[34] and scanning transmission electron microscopy (STEM)[7] have also been performed to achieve the in situ monitoring of the in-plane relative gliding and dynamic phase boundary movement during ion embedding. Furthermore, the relevant first-principles calculations based on density function theory (DFT) have shown that the lattice parameter "$a$" increases from 3.147 Å (MoS$_2$) to 3.252 Å (Li$_x$MoS$_2$) with a corresponding compressive strain of 3.33%[40]. The eigen flexibility property[40] of MoS$_2$ helps such strain and stress to produce a biaxial compressive force in the plane of materials after constraint by the hard substrate, resulting in distortion by microstrains. These processes are accompanied by the formation and growth of the planar nanofolds, and the monolayer MoS$_2$ finally develops into a dense wrinkle-structure network[41].

In addition, it should be realized that the formation of wrinkle structures upon lithiation might also require either deformation of the underlying material or debonding from the underlying material. To clarify this point, in situ AFM imaging on the Si electrode in the same electrolyte is shown in Supplementary Fig. 5, which indicates that no obvious deformation occurs on Si during the potential range of wrinkle formation on MoS$_2$ from OCP (2.64 V) to cathodic 1 V. It is accordingly revealed that the Si electrode itself is unable to generate deformation to induce wrinkling of the MoS$_2$ layer upon lithiation. Additionally, the

related DMT modulus measurement of wrinkle-structure networks on monolayer $MoS_2$ is shown in Supplementary Fig. 6, indicating that the underlying Si substrate with a higher modulus has almost no distinct deformation upon lithiation. Consequently, the observed wrinkle formation does not require deformation of the underlying Si substrate.

On the other hand, understanding the pathway of $Li^+$ migration can further help us to estimate whether the proposed wrinkling requires debonding from the underlying material. Clearly, in regard to the migration path of $Li^+$ in the case of monolayer $MoS_2$ on the Si substrate, two possibilities can be proposed. One is that $Li^+$ directly adsorbs on the surface of $MoS_2$, and the other is that $Li^+$ interposes between the $MoS_2$ and substrate from the edge sites of the monolayer $MoS_2$. First, if a majority of $Li^+$ ions directly adsorb on the electrode, an even germination of wrinkles will occur. For another, if $Li^+$ intercalation mostly occurs between the $MoS_2$ and substrate, the growth of wrinkle-structure networks will obviously exhibit an edge-to-centre mode. The obtained in situ AFM results demonstrate that the wrinkle structure is uniformly generated on the electrode surface upon lithiation instead of showing an edge-to-centre growth mode. Accordingly, it could be considered that the majority of $Li^+$ ions prefer to directly adsorb on the monolayer $MoS_2$ surface for further reaction with $MoS_2$ rather than intercalate into the intermediate layer at the given potential range. On this basis, it is indicated that the wrinkling in the present system likewise does not require debonding from the underlying material.

**Quantitative live formation of ultra-thin FEC-derived SEI**. To further explore the entire process of SEI formation and the surface effect of the electrolyte additive in situ and quantitatively, a 10 wt% FEC additive was added to the electrolyte to execute operando EC-AFM characterization. An atomically flat platform of $MoS_2$ without any impurities is exhibited at OCP (Fig. 2a). Upon charging to 1.77 V, a brush-shaped film is initially generated on the $MoS_2$ surface, as indicated by the yellow arrows (Fig. 2b). Interestingly, the next scanned plot shows that this initial start to lateral growth develops into an intensive and uniform film covering the majority of the electrode, as shown within the yellow dotted line in Fig. 2c. Remarkably, the dense and homogeneous coverage of such an ultra-thin film above the whole $MoS_2$ surface cannot be observed in the absence of FEC, and thus called FEC-derived SEI film (Fig. 2d). The dynamic nucleation and growth processes are shown in more detail in Supplementary Movie 1, in which the interfacial morphology remains invariable from OCP to cathodic 1.94 V, whereas the abrupt interphasial formation of the ultra-thin FEC-derived SEI film is captured kinetically (the scale bar is 600 nm in the video). The thickness is quantitatively measured as approximately 0.6 nm during the SEI film-forming nucleation and initial growth processes, and the specific AFM height section profile along the yellow dashed line of the entirely covered FEC-derived SEI film in Fig. 2d is shown in Fig. 2d'. The quantitatively measured thicknesses of FEC-derived SEI film at a certain position during the whole charging process is shown in Fig. 2j (specific measurement statistics as indicated in Supplementary Table 1). Such ultra-thin FEC-derived SEI film nucleates at the interface with a thickness of $0.7 \pm 0.1$ nm and then substantially maintains the nearby height by propagating to the total electrode surface (indicated as the $j_1$ process). The film gradually accumulates thickness but nevertheless still retains an average thickness of approximately $1.5 \pm 0.7$ nm upon lithiation (shown as the $j_2$ process). During in situ AFM cycles, the thicknesses of FEC-derived SEI film ranging from 1.4 to 3.1 nm with an average value of $2.4 \pm 0.2$ nm shows no

conspicuous variation (detailed AFM images, thickness-cycle number graph and corresponding statistics are shown in Supplementary Figs. 7 and 8 and Table 2). These results adequately reveal the ultra-thin nature and interphasial homogeneity of the FEC-derived SEI film upon cycles. It has been reported that a dense and uniform SEI film with fewer localized defects simultaneously possesses an organic–inorganic bilayer structure and appropriate thickness can usually reduce the inhomogeneous reaction and volume expansion of the electrode by balancing the ion transport and interface stabilization[12,16,49]. Based on the relevant studies, in our experiments, dense and uniform coverage of the ultra-thin FEC-derived SEI film is favourable to effectively prevent side reactions in the electrode, significantly improving battery performance (Fig. 2k). The subsequent growth and accumulation of highlighted NPs arise at the $MoS_2$/electrolyte interface during subsequent charging to 1.22 V, establishing a special double-layer structure of FEC-derived and NP-shaped SEI films (Fig. 2e), respectively marked as SEI-I and SEI-II in Fig. 2f (magnified view of the yellow box in Fig. 2e). Notably, the FEC-derived SEI film generates earlier than NP-shaped one, which is supported by the previous finding[50] that additives undergo prior reduction to construct a preliminary SEI film for pre-protection upon electrode reactions[50]. The discrepant dynamic behaviours of the NP-shaped and FEC-derived films upon cycling further distinctly elucidate the differences in essence and properties of the two distinguishable SEI films (thickness–cycle number graph and corresponding statistics are respectively shown in Supplementary Fig. 9 and Table 3). Such a large-area ultra-flat monolayer $MoS_2$ in FEC-containing electrolyte offers an extremely suitable substrate and favourable system for the in situ visualization of the nanoscale nucleation and growth processes of two diacritical SEI films, providing more evidence and details on the quantitative live formation of ultra-thin SEI films and the surface effect of electrolyte additives, and is admittedly of significant interest to studies of SEI films in batteries.

As $Li^+$ intercalation proceeds, the bulges and nanofolds gradually evolve into wrinkling networks owing to the release of in-plane compressive stresses (Fig. 2g, h). The wrinkle structure is similar to but sparser than that in the absence of FEC (Fig. 1h, i), suggesting that this bilayer structure of SEI film is effective in relieving stress generation and further suppressing volume expansion, and the extra-developed ultra-thin FEC-derived SEI film should be chiefly responsible. The planar network nanostructures remain at the interface after delithiation to anodic 3.5 V (Fig. 2i), revealing the partial reversibility of the first cycle in $MoS_2$-based LIBs. Supplementary Movie 2 directly and distinctly shows the dynamic lithiation/delithiation processes at the $MoS_2$/electrolyte interface, which is in situ characterized by Fastscan AFM, whose adopted scan rate is accordingly three times that of ordinary AFM (the scale bar is 600 nm in the video). The interfacial evolution of wrinkling formation and shrinkage during charging/discharging processes is apparently captured with a higher temporal resolution, and such an ultra-thin FEC-derived SEI film is still uniformly preserved at the interface after discharging, further demonstrating the favourable interphasial structure and function of such SEI originating from film-forming additives. The variation in average roughness in AFM images is additionally used as a semi-quantitative insight into the reversibility and capacity attenuation of batteries upon charging/discharging. The average roughness of AFM images after lithiation and delithiation in the FEC-containing system is 5.7 (Fig. 2h) and 5.2 nm (Fig. 2i), and the corresponding measurements in the FEC-free system are 3.7 nm (Supplementary Fig. 4b) and 3.5 nm (Supplementary Fig. 4c), and the average roughness declines by 8.77% and 5.41% after delithiation in the FEC-containing and FEC-free systems, respectively. The quantitatively

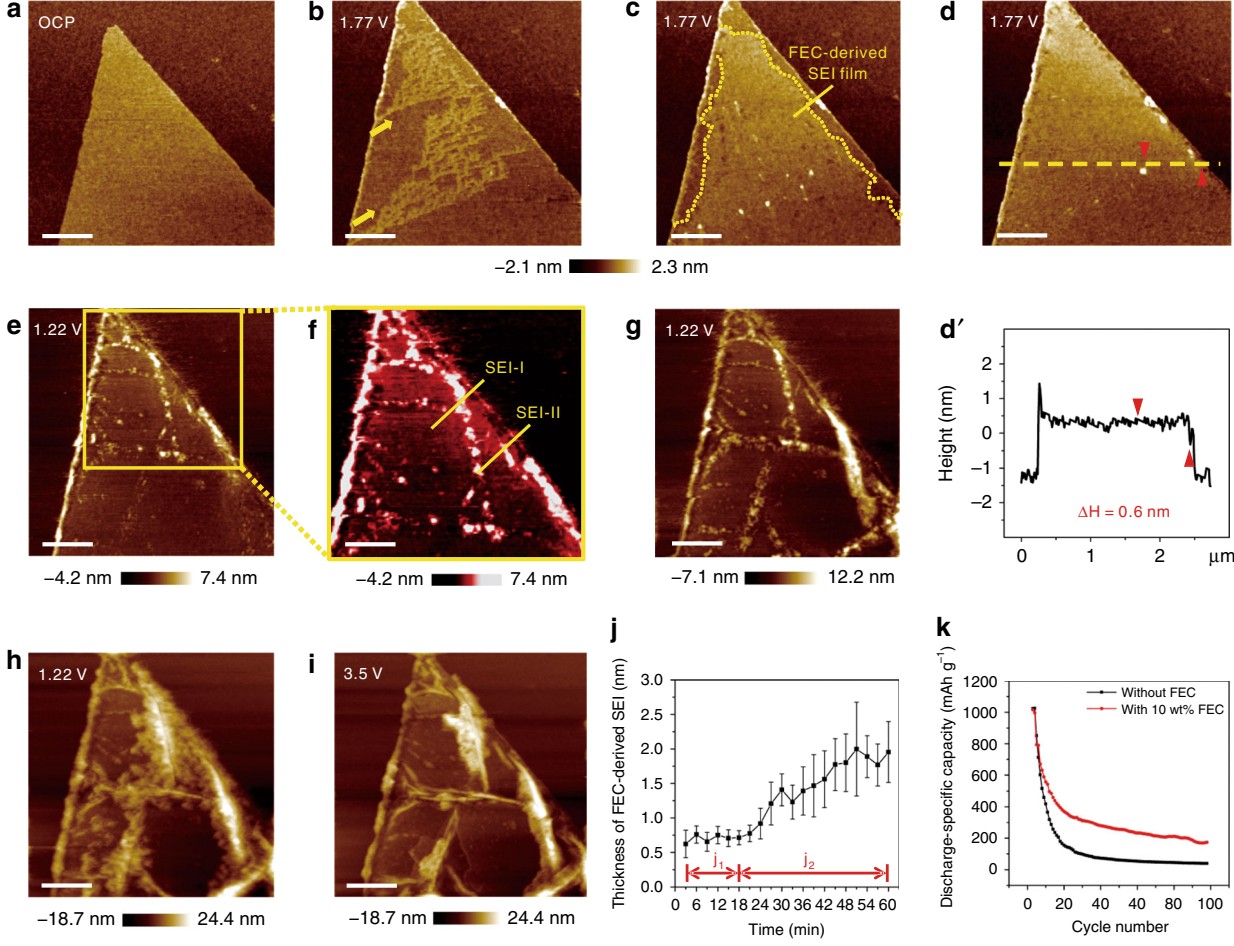

**Fig. 2** In situ and quantitative insights into the live formation of ultra-thin FEC-derived SEI films. AFM images of monolayer $MoS_2$ in $[BMP]^+[FSI]^-$ containing 0.5 M LiFSI electrolyte with 10 wt% FEC additive at different potentials of **a** OCP, cathodic **b–d** 1.77 V, **e–h** 1.22 V and **i** anodic 3.5 V. **d′** The thickness of such ultra-thin FEC-derived SEI film is 0.6 nm from the AFM height section profile along the yellow dashed line in **d**. **f** Magnified view of the yellow box in **e**, which is differentiated with a strong contrast. The interphasial bilayer structure is marked as SEI-I (FEC-derived SEI) and SEI-II (NP-shaped SEI). The scale bar is 600 nm in **a–i**. **j** Quantitatively measured thickness of FEC-derived SEI film at a certain position upon charging, including (j₁) SEI formation and (j₂) lithiation processes. **k** Cycling performance of Li/$MoS_2$ coin cells in FEC-free and FEC-containing systems at 0.05 A g⁻¹

measured heights of wrinkles in the two systems upon cycling further demonstrate a sparser and more stable wrinkling network in the electrolyte with 10 wt% FEC additive (detailed AFM images of FEC-containing/free systems are shown in Supplementary Figs. 7 and 10; the thickness–cycle number graph and corresponding statistics are given in Supplementary Fig. 11 and Table 4). These results show that the preliminary ultra-thin FEC-derived SEI film plays a crucial role in optimizing the stability of electrode/ electrolyte interfaces and further improving the reversibility and capacity of batteries during cycles. The results above achieve a comprehensive glimpse into the functional live formation of additive-derived SEI film and the corresponding degradation mechanism upon charging/discharging, which could furnish more predictive evaluations and constructions of SEI film quality and battery performance.

**Defective effect on surface dynamics.** It is well known that not only the electrolyte additives but also the electrode structures greatly affect the electrochemical performances of batteries. Multilayer $MoS_2$ with a high density of step edges was further employed to study the impact of localized defects on the interfacial evolution and reaction dynamics via in situ EC-AFM. The topological structure of a multilayer-triangular $MoS_2$ with a layer spacing of approximately 0.7 nm at OCP (~2.93 V) is shown in

Fig. 3a. The step edges of the multilayer $MoS_2$ electrode are clearly highlighted (Fig. 3b) due to the adsorption of reduction products of $[FSI]^-$ in the electrolyte at cathodic 1.78 V (accurate measurement of the interlayer spacing is shown in Supplementary Fig. 12), which is consistent with the phenomena exhibited in Fig. 1d. The electrode/electrolyte interface remains unchanged from cathodic 1.78 to 1.29 V and an AFM image at 1.29 V is shown in Fig. 3c. The interfacial structure evolves suddenly and rapidly from triangles to wrinkle-structure networks at the negative shift from 1.09 to 1.01 V (Fig. 3d), fully revealing the intrinsic flexibility of $MoS_2$ materials. Moreover, the gradient of average roughness in AFM images is further used as a semi-quantitative study on the speed of reaction dynamics. The average roughness of AFM images before and upon lithiation in monolayer $MoS_2$ system is measured as 1.2 nm (Fig. 1f) and 2.9 nm (Fig. 1g), and that of multilayer $MoS_2$ is 2.4 nm (Fig. 3c) and 5.7 nm (Fig. 3d). During the same scan time (~3 min), the gradient of average roughness in the multilayer $MoS_2$ system is twice that of monolayer $MoS_2$, suggesting that the presence of localized defects contributes to the embedding of Li⁺ and therefore enhances the reaction kinetics. as lithiation proceeds, wrinkles appear on the bottom monolayer $MoS_2$ (Fig. 3e) and then branch and spread from defects to the surrounding regions (Fig. 3f), which are marked by yellow arrows. More defective edges and a relatively

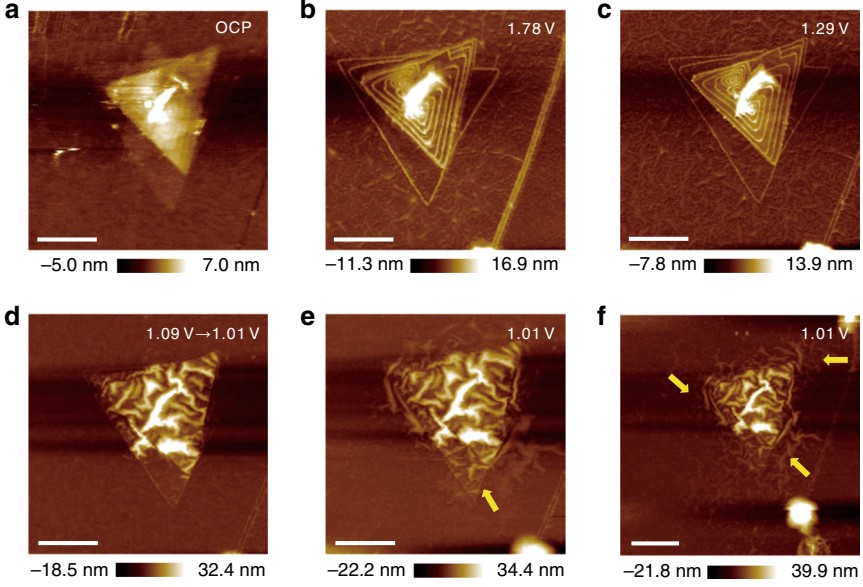

**Fig. 3** In situ monitoring of the structural evolution on a multilayer MoS$_2$ electrode. AFM images of the interfacial morphologies in [BMP]$^+$[FSI]$^-$-containing 0.5 M LiFSI electrolyte at different potentials of **a** OCP, cathodic **b** 1.78 V, **c** 1.29 V, **d** 1.09 to 1.01 V, **e**, **f** 1.01 V. The scale bars are 500 nm in **a–e** and 600 nm in **f**

shorter Li$^+$ migration path should be responsible for the faster lithiation dynamics of the multilayer MoS$_2$ electrode, and corresponding 3D AFM images of the monolayer and multilayer MoS$_2$/electrolyte interfaces after lithiation are shown in Supplementary Fig. 13 for a space comparison. After discharging to 2.98 V, the interface remains uncontracted wrinkles, and the volume expansion cannot be completely reversible (Supplementary Fig. 14), elucidating the inherent irreversibility and capacity attenuation of MoS$_2$ electrodes. These phenomena microscopically reveal the effects of surface defects on the reaction kinetics, which helps to more deeply understand the structure–reactivity correlations.

**Characterization of SEI film and phase transition**. To clarify the evolution of the chemical component during the electrochemical process, further ex situ characterizations were conducted to compare the NP-shaped and FEC-derived SEI films and the phase transition on the MoS$_2$ electrode. The F 1s XPS spectra of the samples after lithiation in the electrolyte with/without 10 wt% FEC (Supplementary Fig. 15) reveal two peaks at approximately 684.7 and 687.5 eV that are attributed to LiF and LiFSI[51], and the peak area ratios of LiF/LiFSI are 0.62 and 2.18 in the FEC-free and FEC-containing systems, respectively. Therefore, the LiF-rich composition of the FEC-derived SEI film can be clearly elucidated and is beneficial to prevent further electrolyte decomposition because of the suppression of side reactions between the electrode and electrolyte owing to the low electrical conductivity of LiF[10,23].

The phase transition of the MoS$_2$ electrode is generally explored using HR-TEM, XPS and Raman spectra. Figure 4a, b shows HR-TEM images of samples at OCP and after lithiation, respectively, and enlarged views of two images are shown as illustrations, where red circles represent the Mo atoms and yellow ones represent the S atoms. The MoS$_2$ electrode at OCP is a trigonal prismatic 2H phase with three S atoms surrounding each Mo atom from the z-axis direction (Fig. 4a); however, there are six S atoms around the Mo atom (the Mo atoms are not shown) in MoS$_2$ after lithiation (Fig. 4b). It has been reported that the implantation of Li$^+$ causes S atoms in the lower layer to rotate by 30°, resulting in the local rearrangement of S atoms and thereby

forming the octahedral 1T phase of Li$_x$MoS$_2$ with six S atoms around each Mo atom[42,52]. Therefore, the phase transition of MoS$_2$ induced by Li$^+$ intercalation could be directly confirmed by HR-TEM. The XPS results are shown in Fig. 4c. The blue and red lines represent the Mo 3d XPS spectra of MoS$_2$ at OCP and after lithiation, respectively. The peaks of the sample at OCP are 230.1 and 233.3 eV, which are ascribed to Mo 3d$_{5/2}$ and Mo 3d$_{3/2}$, respectively, and the corresponding two peaks in the sample after lithiation are reduced by 0.9 and 0.5 eV, which is consistent with the attributable peak position of the 1T phase[53]. The Raman spectrum is also used to further verify the phase transition upon lithiation. The blue and red lines in Fig. 4d exhibit the Raman shifts of samples at OCP and after lithiation. The characteristic peaks of 2H MoS$_2$ are 282, 380, 400 and 451 cm$^{-1}$, representing the E$_{1g}$, E$_{2g}^1$, A$_{1g}$ and longitudinal acoustic phonon modes, respectively[54]. Additional peaks at 195, 230 and 335 cm$^{-1}$ observed in the sample after lithiation are attributed to the three phonon modes of 1T phase[55,56]. Thus, the phase transition from 2H MoS$_2$ to 1T Li$_x$MoS$_2$ caused by Li$^+$ intercalation is fully proven.

**Structural evolution and reaction mechanism at the interface**. Based on the above experimental results, the interfacial mechanism of electrochemical behaviours upon charging/discharging is proposed (Fig. 5). Figure 5a presents a large-area ultra-flat monolayer MoS$_2$ on an Si substrate at OCP. In the FEC-free system, bright NPs appear at the edge and platform of the MoS$_2$ electrode as the interphasial nucleation (Fig. 5b), and the sustained growth and accumulation of NPs form an NP-shaped SEI, which is dispersed on the MoS$_2$ (considering the 3D and sectional views in Fig. 5c). With the abundant intercalation of Li$^+$, a phase transition from 2H (trigonal prism) to 1T (octahedron) occurs on the MoS$_2$ electrode. Three S atoms are arranged around each Mo atom in 2H MoS$_2$ along the z-axis direction (Fig. 5j), whereas the S atoms in the lower layer rotate by 30° so that each Mo atom is enclosed by six S atoms after lithiation (Fig. 5k)[41,53]. Furthermore, stresses are generated upon the phase transition and finally released by producing a wrinkle-structure network (Fig. 5d). The planar nanofolds still remain at the interface after

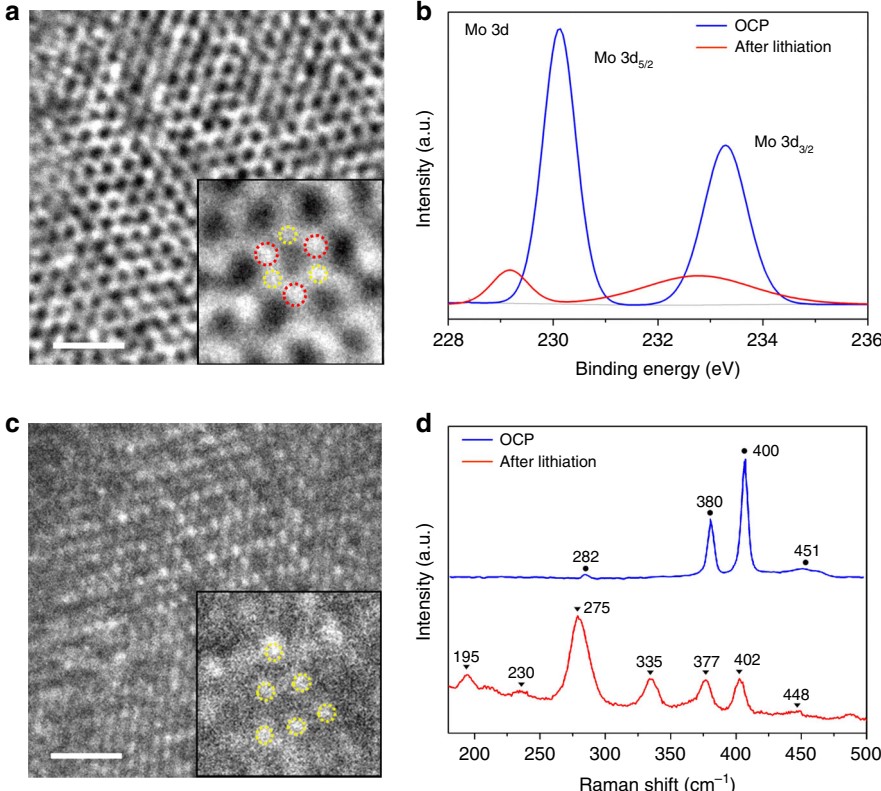

**Fig. 4** Ex situ characterizations of the phase transition on the MoS₂ electrode upon lithiation. HR-TEM images of the MoS₂ specimen obtained **a** at OCP and **b** after lithiation. Insets show the enlarged views, where red circles represent Mo atoms and yellow ones represent S atoms. The scale bar is 1 nm in **a**, **b**. **c** Mo 3d XPS and **d** Raman spectra of the samples at OCP and after lithiation, indicated as blue and red lines, respectively

delithiation (Fig. 5e), which ultimately leads to the efficiency decline and capacity attenuation of batteries in the first cycle.

In contrast, a dense, uniform and ultra-thin FEC-derived SEI film with a thickness of several nanometres nucleates (Fig. 5f) and grows (Fig. 5g) at the MoS₂/electrolyte interface in the FEC-containing system. The FEC-derived SEI is generated earlier than the NP-shaped SEI, revealing the preliminary protection of the film-forming additive, developing an interphasial double-layer structure (Fig. 5g). Wrinkling nanostructures appear on the electrode due to stresses released during lithiation, which are similar to but sparser than those in the FEC-free system (Fig. 5h), and the relatively significant volume shrinkage of MoS₂ exhibited after delithiation demonstrates a smaller capacity loss in the FEC-containing system (Fig. 5i). These investigations reveal the interphasial properties and functions of ultra-thin FEC-derived SEI films on the enhancements of electrode toughness and internal stress reduction, offering further optimized strategies of reversibility and stability improvements.

## Discussion

In summary, in situ and quantitative EC-AFM was successfully conducted to directly and meticulously investigate the interfacial evolution and reaction mechanism of SEI film live formation and dynamic lithiation/delithiation on the large-area ultra-flat monolayer MoS₂ electrode. An ultra-thin FEC-derived SEI and an NP-shaped SEI are distinguished to develop a special interphasial bilayer structure. This ultra-thin SEI film initially nucleates and spreads to the entire electrode with a substantial thickness of approximately $0.7 \pm 0.1$ nm and then accumulates to $1.5 \pm 0.7$ nm thick, fully revealing the ultra-thin nature and interphasial homogeneity of the FEC-derived SEI film. The reduction in side reactions and provision of preliminary protections for the MoS₂

electrode is further elucidated by the LiF-rich and earliest generated properties of such ultra-thin additive-derived SEI films. The intrinsic flexibility of MoS₂ is expounded by wrinkles owing to the phase transition upon lithiation, and the maintenance of wrinkling after delithiation reveals the capacity degradation mechanism. These results provide straightforward evidence for the quantitative live formation and interphasial morphology of ultra-thin SEI films and further reveal the underlying surface effect of additives and structure–reactivity correlations of MoS₂-based LIBs from in-depth insights into the interfacial evolution and corresponding reaction mechanism. A deep understanding of these fundamental functional SEI properties and interfacial dynamic attenuation will yield information on controllable interfacial engineering and prospective optimized designs in the present and future battery community.

## Methods

**Large-area ultra-flat monolayer MoS₂ CVD growth**. MoS₂ monolayers were grown by atmospheric pressure CVD. The growth substrate (SiO₂/Si) was cleaned in acetone, isopropanol, and deionized water and then dried under high-purity N₂ flow. For the MoS₂ monolayers, the growth substrate was placed face-down above a ceramic boat containing MoO₃ powder (20 mg). The ceramic boat was then loaded in the central heating zone of the furnace tube, where another boat containing 80 mg of sulfur was located upstream. The tube was first purged with ultrahigh-purity Ar for 10 min at a flow rate of 200 sccm. Then, the furnace was heated from room temperature to 500 °C in 12 min, heated from 500 to 720 °C in another 20 min, and then maintained at 720 °C for 5 min. The sulfur was heated to 130 °C with a separate heat belt as the furnace reached 720 °C. Finally, the temperature was cooled from 720 to 570 °C in 20 min before opening the furnace for rapid cooling. Ar as carrier gas was maintained at a flow rate of 10 sccm throughout the growth process.

**MoS₂ monolayer transfer**. First, a poly(methyl methacrylate) (PMMA) thin film was spin-coated on top of the MoS₂/SiO₂/Si substrate. After that, the SiO₂ layer was etched by 2 M KOH solution, and the PMMA/MoS₂ layer was lifted off. The

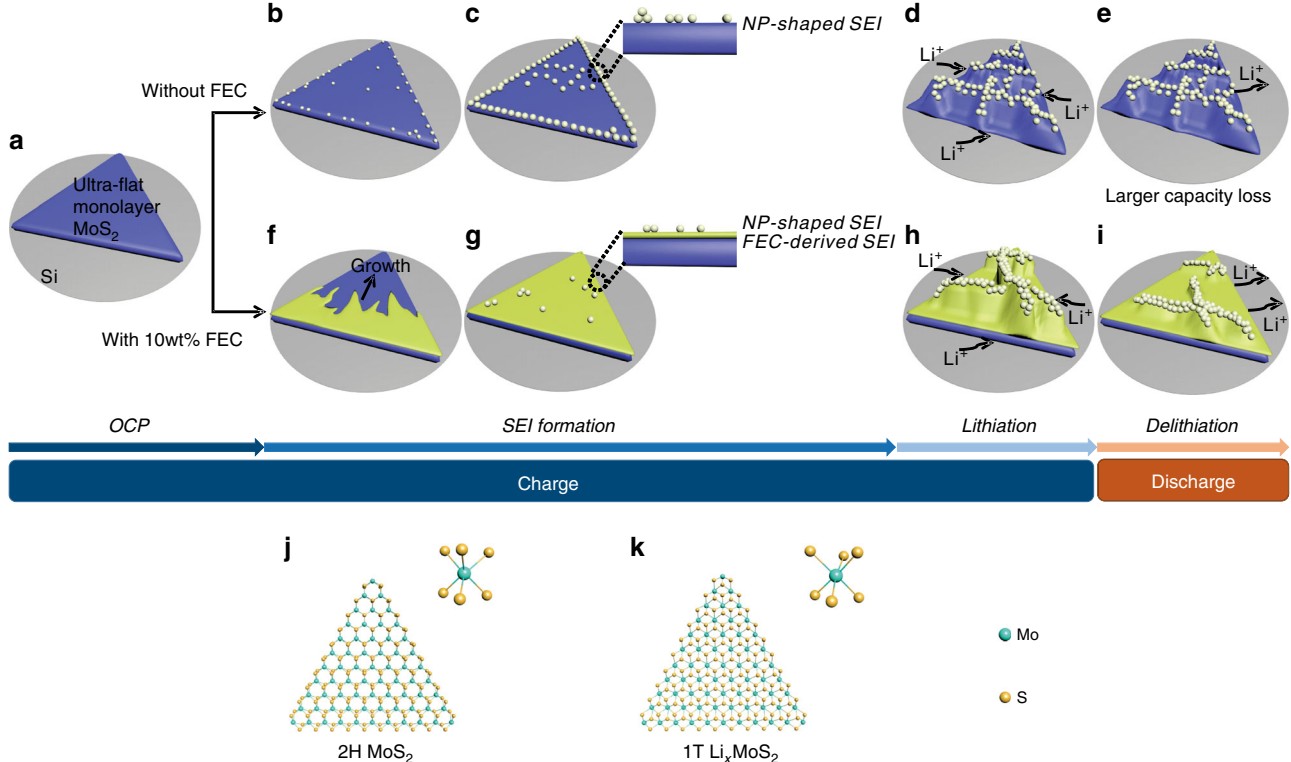

**Fig. 5** Schematic illustration of the structural evolution and reaction mechanism of the monolayer $MoS_2$ electrode. Interphasial formation and lithiation/dilithiation at the interface in **a–e** FEC-free and **a, f–i** FEC-containing systems. **a** An ultra-flat monolayer $MoS_2$ with an atomically flat terrace on an Si substrate is shown at OCP. **b** NP nuclei appear at the edge and platform. **c** The growth and accumulation of NPs generate an NP-shaped SEI on $MoS_2$. **d** Reticular wrinkles arise with the intercalation of $Li^+$. **e** Wrinkles irreversibly remain after delithiation, indicating the failure mechanism. **f** The initial formation of FEC-derived SEI film. **g** Ultra-thin FEC-derived SEI film covers the entire $MoS_2$ electrode densely and uniformly with bits of NPs adsorbed, developing an interphasial bilayer SEI structure. **h** Sparse wrinkles form upon lithiation. **i** More apparent volume shrinkage of the electrode compared with the FEC-free system after delithiation, manifesting a smaller capacity loss with 10 wt% FEC additive. **j**, **k** Phase transition process caused by $Li^+$ implantation. **j** Trigonal prismatic 2H $MoS_2$ before lithiation, where three S atoms are arranged around each Mo atom. **k** Octahedral 1T $Li_xMoS_2$ with the embedding of $Li^+$, where six S atoms are arranged around each Mo atom. Green and yellow spheres represent Mo and S atoms, respectively

PMMA/$MoS_2$ was then transferred onto the Si (100) substrate and air-dried. PMMA was subsequently washed off with acetone and 2-propanol.

**In situ electrochemical AFM/Fastscan AFM experiments**. A three-electrode system was adopted in the electrochemical experiments. Fabricated and transferred monolayer/multilayer $MoS_2$ electrodes were used as the working electrode, and the counter and reference electrodes were lithium wires. Three electrodes were enclosed into a homemade electrochemical cell in the glove box (Mikrouna, Super 1220/750, $H_2O < 0.1$ ppm, $O_2 < 0.1$ ppm) filled with high-purity Ar, and an "O" ring with a diameter of 8 mm was used to seal the cell. The electrolytes were $[BMP]^+[FSI]^-$ IL containing $0.5$ mol $L^{-1}$ LiFSI with/without 10 wt% FEC, in which $[BMP]^+[FSI]^-$ and LiFSI were from TCI Corp., and FEC was purchased from Sigma-Aldrich.

In situ EC-AFM/Fastscan AFM experiments were carried out in the Ar-filled glove box by combining AFM (Bruker Corp., Dimension Icon)/Fastscan AFM (Bruker Corp., Dimension Fastscan) with an electrochemical workstation (Methrohm Autolab, PGSTAT302N). All potentials were referred to $Li^+$/Li. The potential was swept towards negative potential from OCP to 1.0 V during the charging process and towards positive potential to 3.5 V for subsequent oxidation at the scan rate of 1 mV $s^{-1}$. AFM images were acquired by using an insulating triangular silicon nitride AFM tip (Bruker Corp., $k = 26$ N $m^{-1}$, $f_0 = 300$ kHz) to scan the surface of $MoS_2$ in the mode of PeakForce QNM (Quantitative Nano Mechanics). For the recording of Fastscan AFM images, a specialized Fastscan-B probe (Bruker Corp., $k = 4$ N $m^{-1}$, $f_0 = 400$ kHz) was applied in the Fastscan mode of ScanAsyst.

Additionally, the $MoS_2$ powders were mixed with super P (conductive additive) and PVDF (binder material) in a ratio of 8:1:1 and then assembled into coin cells with $[BMP]^+[FSI]^-$ containing 0.5 M LiFSI with/without 10 wt% FEC as the electrolytes for electrochemical testing. The cycling performances of Li/$MoS_2$ coin cells in FEC-free and FEC-containing systems were obtained at a current density of 0.05 A $g^{-1}$.

**Ex situ characterizations**. SEM (Hitachi S-4800) and optical microscopy (Nikon Eclipse LV100D) were employed to image the samples of large-area ultra-flat monolayer $MoS_2$ fabricated by the CVD method. $MoS_2$ electrodes were separated from respective coin cells via constant potential control at specific stages and were rinsed with dimethyl carbonate (DMC, Sigma-Aldrich) solution to remove the residual electrolyte on the surface. They were dried in an Ar-filled glove box before ex situ characterizations. The HR-TEM images were obtained by a TEM (JEM 2100F, JEOL, Japan) with an accelerating voltage of 200 kV. Raman spectroscopy (Thermo Scientific DXR, 532 nm laser wavelength) and XPS were also performed to further demonstrate the phase transition upon $MoS_2$. The XPS measurements were conducted on an ESCALab220i-XL electron spectrometer (VG Scientific) using 300 W Al $K_\alpha$ radiation ($hv = 1486.6$ eV). The base pressure was approximately $3 \times 10^{-9}$ mbar, and the binding energies were referenced to the hydrocarbon C 1s peak at 284.8 eV.

## Data availability

All data supporting this study and its findings are available within the article and Supplementary Information. Additional supporting data of this study are available from the corresponding author on reasonable request.

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

## Acknowledgements

We thank Ms. Hui-Ning Li, Institute of Chemistry, Chinese Academy of Sciences for the HR-TEM measurements. This work was financially supported by the National Key R&D Program of China (Grant No. 2016YFA0202500), National Nature Science Fund for Excellent Young Scholars (Grant No. 21722508), National Natural Science Foundation (Grant No. 21573253), "Hundred Talents Program" from Chinese Academy of Science and 111 Project (No. B12015) from Key Laboratory of Advanced Energy Materials Chemistry, Nankai University.

## Author contributions

J.W. performed the in situ AFM experiments and the Raman characterizations and analysed the data. Y.H. synthesized the large-area monolayer $MoS_2$. Y.S. and H.-J.Y. assisted in the in situ AFM studies. Y.-X.S. assisted in the Raman measurements. J.Z. and R.W. conceived the study and supervised the research with the help of L.-J.W. J.W. and R.W. wrote the manuscript. All authors discussed the results and commented on the manuscript.

## Additional information

**Competing interests:** The authors declare no competing interests.

