## [Peer Review File · Nature Communications]

Reviewers' comments:

Reviewer #1 (Remarks to the Author):

The results reported here are interesting, and likely to be of interest to the battery community. I believe that the following improvements are needed prior to publication:

1. How do the authors know that they are producing a monolayer of MoS₂? This is not clearly described.
2. I do not fully understand how the authors are distinguishing surface features from wrinkles. This should be explained more clearly.
3. The authors should reconsider their description of stress generation more carefully. I am not all that familiar with work on MoS₂. However, by analogy with graphite, Li ions insert between the 2D layers, such that most of the volume expansion occurs along the c-axis. For a thin film this means that the volume expansion will occur normal to the substrate, and not lead to significant stress in the plane of the film. In contrast to this, the expected out-of-plane strain should not lead to wrinkling.
4. In a monolayer film, it is not clear where the Li is going. Describing this issue more directly should help clarify issues that are related to the source of the stress.
5. As an alternative, it is also possible that stress is created inside of the SEI itself, as it forms. This has been reported in graphite (see Tokranov et al, Journal of the Electrochemical Society 161, A58-A65 (2014)).
6. The authors present the following interpretation: "Such ultra-thin FEC-derived SEI does not affect the migration of Li⁺, but helps to reduce side reactions at the electrode/electrolyte interface and prevent further decomposition of the electrolyte because of its dense and uniform coverage on the electrode surface, which contributes to the improvement of battery performances (Fig. 2k)." What is this statement based on? It strikes me as rather speculative. The authors should explain their reasoning more carefully.
7. I am surprised that the authors do not cite Doron Aurbach's seminal work on the use of AFM to study SEI.
8. Some improvements in the English language are needed. While the authors' meaning can be understood in most places, there are also a number of places where more clarity is needed.

Reviewer #2 (Remarks to the Author):

This very interesting manuscript describes SEI formation and growth for lithiation and delithiation of a monolayer of MoS₂ in presence of two types of liquid electrolytes. The nature of the ML MS₂ makes the system ideal for this type of study and certainly the various stages are very well analyzed and discussed. The results are extremely useful for understanding of the SEI not only on this material but also in others. I suggest that before acceptance, the authors comment on the following aspects of their work.

1. The authors claim that the ultra-thin FEC film helps controlling the reactivity and therefore the

number of reduction reactions in successive cycles tends to decrease. During delithiation the discussion after Figure 2 indicates that both film thickness and height of the NPs are reduced. Are these SEIs partially dissolved in the electrolyte? Or is it a reorganization of the structure of both the LiF film and the NPs? Please explain and show any evidence of either phenomena.

2. How does the thin film thickness (LiF) grow in successive cycling?

3. How does the phase transition affect the SEI formation if at all? Does it affect lithiation or delithiation? How?

4. Do the wrinkles grow upon cycling? How many cycles could this type of electrode be used for each electrolyte?

Reviewer #1 (Remarks to the Author):

The results reported here are interesting, and likely to be of interest to the battery community. I believe that the following improvements are needed prior to publication:

1. How do the authors know that they are producing a monolayer of MoS₂? This is not clearly described.

Response: We thank the reviewer for the comments. In the present work, the ultra-flat monolayer MoS₂ electrode was synthesized and fabricated by chemical vapor deposition (CVD). Researches on the preparation and growth of monolayer MoS₂ via using the CVD method has been previously reported¹. And in order to acquire the large single-crystal and high-quality monolayer MoS₂ by CVD, optimizing strategies, such as oxygen assistance² and seeding promoter³, have also been further explored and innovated.

After obtaining the produced and transferred monolayer MoS₂ electrode, atomic force microscopy (AFM) measurement was firstly performed. The thickness of such MoS₂ electrode is 0.7 ± 0.2 nm from the AFM cross-section profile along the dashed line indicated in Supplementary Figure S2a, manifesting the nature and essence of monolayer MoS₂.

The AFM and corresponding height section images are shown in the Supplementary Figure S2 below, and this part has been updated accordingly in Supplementary Information. And the description of such monolayer MoS₂ electrode has further been added in highlights of the page 6 line 15 of the revised manuscript, as “Fig. 1c shows AFM image of monolayer MoS₂ electrode at open circuit potential (OCP, ~ 2.64 V), where atomically flat terrace (0.7 ± 0.2 nm in thickness and specific height section profile is shown in Supplementary Fig. S2) can be clearly captured.”.

Supplementary Figure S2. **The AFM image and height section of the monolayer MoS₂ electrode.** (a) AFM image of the MoS₂ electrode after be produced by CVD and transferred to Si substrate. The scale bar is 600 nm. (b) The cross-section profile along the dashed line indicated in (a).

2. I do not fully understand how the authors are distinguishing surface features from wrinkles. This should be explained more clearly.

Response: We thank the reviewer for comments and suggestions. The microscopic morphology, wrinkle-structure networks, of lithium-intercalated MoS₂ has been beforehand followed by scanning electron microscopy (SEM) and laser confocal microscope combined with a differential interference contrast microscope (LCM-DIM)^{4,5}. And similarly wrinkling behavior of MoS₂ anode caused by the embedding of Na ions has also been monitored and captured by AFM observations in sodium-ion batteries (NIBs)⁶.

In our experiments, the *in situ* AFM characterizations provide insights into the growth and development of mesh-distributed wrinkles, as exhibited in Figure 1g-i and Figure 2g-h, detailed description is accordingly indicated in the main text of manuscript. 3D AFM images of wrinkling appearance of monolayer and multilayer MoS₂ electrode are also shown in Supplementary Figure S11, and the dynamic growth processes of wrinkle-structure networks can be discovered in Supplementary Video 2.

Furthermore, the 2D and 3D AFM images of wrinkles morphology in other repeated experiments have been additionally provided in Supplementary Figure S3, where the ravines and valleys of nanofolds can be clearly distinguished as the surface features from wrinkles. The corresponding expression has been added in highlights of the page 7 line 19 of the revised manuscript, as “Detailed ravines and valleys of mesh-distributed nanofolds are distinctly distinguished as surface features from wrinkles in the 3D AFM images, as shown in Supplementary Fig. S3.”.

Supplementary Figure S3. ***In situ* 2D and corresponding 3D AFM images of the interfacial evolution of wrinkle-structure networks upon lithiation.** 2D AFM images of (a) the initial formation of planar nanofolds at the early stage of lithiation, (b, c) the growth and propagation of interfacial wrinkling nanostructures, and (d) the final appearance of wrinkle-structure networks. Corresponding 3D AFM images (e-h) further manifest the

live-formation of network-distributed wrinkles. The apparent variations of nanofolds are indicated by the yellow arrows in (b-d) and circles in (f-h). The scale bar is 600 nm in (a-d).

3. The authors should reconsider their description of stress generation more carefully. I am not all that familiar with work on MoS₂. However, by analogy with graphite, Li ions insert between the 2D layers, such that most of the volume expansion occurs along the c-axis. For a thin film this means that the volume expansion will occur normal to the substrate, and not lead to significant stress in the plane of the film. In contrast to this, the expected out-of-plane strain should not lead to wrinkling.

Response: We thank the reviewer for the comments and suggestions. Actually, when Li ions only intercalate into the layers of 2D transition metal dichalcogenides (TMDs), the c-axis direction will inevitably produce a certain degree of stress and volume expansion, and such anticipative out-of-plane strain would not lead to wrinkling behavior.

In contrast, the observed wrinkle-structure network formed during lithiation is intrinsically caused by the in-plane compressive stresses of MoS₂ due to the ion implantation^{4,7}. When Li ions embed into the electrode, charge transfer occurs from the intercalant to MoS₂, which will induce the changes in the lattice parameters of lithium-intercalated MoS₂ and the in-plane gliding of S plane, thus further generate the well-known phase transition from 2H-MoS₂ to 1T-Li_xMoS₂.⁸⁻¹⁰ The relevant first-principles calculations based on density function theory (DFT) have manifested that the lattice parameter "a" increases from 3.147 Å (MoS₂) to 3.252 Å (Li_xMoS₂) with a corresponding compressive strain of 3.33%.⁴ Such strain and stress will produce a biaxial compressive force in the plane of materials after being constrained by the hard substrate, resulting in the distortion by microstrains, and subsequently accompany with the formation and growth of the planar nanofolds and finally develop to a dense wrinkle-structure network⁷. Therefore, the internal stress of the MoS₂ material due to changes in the lattice parameters can be commendably relaxed by forming mesh-distributed wrinkles.

Accordingly, we have added the detailed description in the highlights of page 7 line 5 of the revised manuscript, as "Regarding the driving force of the wrinkles, it has been reported that the in-plane relative gliding and dynamic phase boundary movement induced by the charge transfer from intercalated Li⁺, which is intrinsically caused by the interaction of electron doping and electron-phonon coupling, finally result in the coordination structure of metal Mo converting from the trigonal prism to octahedron, thus generate the well-known 2H-1T phase transition to reduce the system energy. Changes in the lattice parameters and structures further produce biaxial compressive stresses and strains in the plane after being constrained by the hard substrate, leading to the distortion by microstrains, and subsequently accompany with the planar nanofolds and ultimately develop to a dense wrinkle-structure network."

4. In a monolayer film, it is not clear where the Li is going. Describing this issue more directly should help clarify issues that are related to the source of the stress.

Response: We appreciate the reviewer for the comments. For a monolayer MoS₂ anode, the migration path of Li⁺ can be generally proposed as two ways. One is that Li ions directly adsorb on the surface of MoS₂, and the other one is that interpose between the electrode and substrate from the edges and defects of the MoS₂. Firstly, if Li⁺ directly adsorbs on the electrode, the stress release due to the phase transition should be in the form of an even germination of wrinkles. On the other hand, if Li⁺ intercalation happens between the MoS₂ and substrate, the growth of wrinkle-structure networks should obviously exhibit as the edge-to-center mode. Based on a large number of *in situ* AFM investigations, we have discovered that planar network nanostructures nucleate, develop and propagate uniformly on the electrode surface, instead of an edge-to-center formation. Typical experimental results of interfacial evolution and reaction dynamics upon wrinkling behavior are detailedly indicated in Figure 1g-i of the manuscript and Supplementary Video 2. In addition, as the response of comment 3 above, surface feature of wrinkles due to the phase transition upon lithiation could be attributed to the in-plane compressive stress, rather than the out-of-plane strain caused by the direct Li⁺ insertion into 2D layers. And the adsorbed Li ions are more favorable to directly react with monolayer MoS₂ electrode. Based on the above analysis, it could be considered that the majority of Li ions prefer to adsorb on the monolayer MoS₂ surface and insert into the tetrahedral coordination center of S-S, thus further lead to the lithiation and phase transition processes, and meanwhile, the possibility can't be ruled out that partial Li⁺ would implant between the electrode and substrate.

In fact, we all realized that there will be a phase transition process from a semiconducting 2H-MoS₂ to metallic 1T-Li_xMoS₂ upon lithiation¹⁰⁻¹⁴. At the early lithiation stage, Li ions implant into the tetrahedral coordination center of the S-S, causing charge transfer from intercalated Li⁺. Due to the co-intercalation of ions and electrons, the coordination structure of metal Mo is converted from the trigonal prism to octahedron, and the corresponding space group is transformed from *P6₃/mmc* to *P-3m1*,¹⁵ thus producing 2H-1T phase transition in order to reduce the energy of the system⁹. Intrinsic semiconductivity of 2H MoS₂ is accordingly changed into metallic 1T Li_xMoS₂, and therefore, a charge density wave (CDW) generates in the 2D plane caused by the appearance of superlattices due to the instability of the low-dimensional metal¹⁶. Associated first-principles calculations elucidate that the essence of the phase transition upon lithiation is the interaction of intrinsic doping and electron-phonon coupling⁸. High resolution transmission electron microscopy (HRTEM)¹⁶ and scanning transmission electron microscopy (STEM)¹⁷ have also been performed to achieve the *in situ* monitoring of the in-plane relative gliding and dynamic phase boundary movement during ion implantation. And the relevant shear mechanism¹⁶ and compressive stress⁴ in 2D planes of MoS₂ have been proposed in phase transition process, that is, the direct causes of wrinkles morphology on MoS₂ electrode. So far, the source and origin of stress during the phase transition caused by lithiation could be clearly explained.

The according interpretation and discussion has been specially added in highlights of the page 7 line 14 of the revised manuscript, as "Based on the *in situ* homogeneous formation (instead of an edge-to-center mode) and the above source and origin of the wrinkles, the migration path of major Li ions can be proposed as the preferential adsorption on the surface of monolayer MoS₂ and then insert into the tetrahedral coordination center of S-S, thus further lead to the lithiation and phase transition processes, and meanwhile, the possibility

can't be ruled out that partial Li^+ would implant between the electrode and substrate.”.

5. As an alternative, it is also possible that stress is created inside of the SEI itself, as it forms. This has been reported in graphite (see Tokranov et al, Journal of the Electrochemical Society 161, A58-A65 (2014)).

Response: We appreciate the reviewer for comments and suggestions. The recommended article (Tokranov et al, Journal of the Electrochemical Society 161, A58-A65 (2014)) is significant for the comprehension of the source and origin of stress upon SEI formation. It has indicated the experimental evidence and corresponding mechanism of the stress evolution during initial SEI formation on graphite. And it has also proposed that the electrode surface is disrupted and amorphized by the limited insertion of solvated ions, and subsequent formation of inorganic SEI suppresses the continuous damage to the electrode. Authors has considered that the relatively soft organic decomposition products and LiF deposition undeniably generate a certain stress during the formation of SEI film, but such amount of stress is not sufficient to compare with the measured compressive stress of -1 GPa. Nevertheless, another more reasonable explanation is that such stress is mainly derived from the embedding of restricted solvated ions¹⁸.

However, for our electrochemical systems, adopted electrolytes are 1-butyl-1-methyl-pyrrolidinium bis (fluorosulfonyl) imide ($[\text{BMP}]^+[\text{FSI}]^-$) ionic liquids (ILs) containing $0.5 \text{ mol}\cdot\text{L}^{-1}$ lithium Bis(fluorosulfonyl)imide (LiFSI) with/without 10 wt% fluoroethylene carbonate (FEC), in which solvated ions do not exist. And in our experiments, there is a slight possibility for FEC-derived SEI film with a thickness of several nanometers and an NP-shaped one of a dozen nanometers to generate the required stress for wrinkles of several tens of nanometers thick. Furthermore, wrinkle-structure networks caused by the stress release occurs at the lithiation potential (~ 1.1 V), whereas the SEI film-forming potentials are higher than lithiation (~ 1.77 V of FEC-derived SEI and ~ 1.39 V of NP-shaped SEI). Additionally, *in situ* AFM experiments can also demonstrate no distinct growth of wrinkles in the nucleation and growth of the two distinguishable SEI films. In summary, we do not exclude the partial stress generation from the SEI films, but the genuine source and origin of planar wrinkle nanostructures is the in-plane compressive stress caused by the intrinsic phase transition which is revealed by the previous responses of comment 3, 4 above.

6. The authors present the following interpretation: “Such ultra-thin FEC-derived SEI does not affect the migration of Li^+ , but helps to reduce side reactions at the electrode/electrolyte interface and prevent further decomposition of the electrolyte because of its dense and uniform coverage on the electrode surface, which contributes to the improvement of battery performances (Fig. 2k).” What is this statement based on? It strikes me as rather speculative. The authors should explain their reasoning more carefully.

Response: We thank the reviewer for the comments and suggestions. As we all know, the

organic SEI film is regarded as the “medium” for Li⁺ transmission, and the inorganic SEI is considered to be responsible for the protection function of the SEI via suppressing the reduction at the SEI/anode interface^{19,20}. Ideally, an organic-inorganic bilayer SEI with an appropriate thickness is beneficial to the ion transport and interface stabilization. Moreover, the dense and uniform SEI film with fewer localized defects, can broadly reduce the inhomogeneous reaction and volume expansion of the electrode²¹. Generally speaking, in our experiments, such an ultra-thin LiF-rich FEC-derived SEI film is capable of balancing ion transport and interface protection. Dense and uniform coverage on the entire surface of the MoS₂ electrode according to the *in situ* AFM investigation, can effectively prevent the continuous decomposition of the electrolyte and the side reaction of the interface, thereby exhibiting a significant improvement in battery performance.

Accordingly, the correlated literature references and explanations have been appended and provided in the highlighted main text of manuscript in page 9 line 21, as “It has been reported that a dense and uniform SEI film with fewer localized defects, simultaneously possesses an organic-inorganic bilayer structure and appropriate thickness, can usually reduce the inhomogeneous reaction and volume expansion of the electrode by balancing the ion transport and interface stabilization. Based on the relevant researches, in our experiments, dense and uniform coverage of the ultra-thin FEC-derived SEI film is favorable to effectively prevent the electrolyte from continuous decomposition and the electrode/electrolyte interface from side reactions, meanwhile without affecting the migration of Li⁺, and thereby exhibiting a significant improvement in battery performances (Fig. 2k).”.

7. I am surprised that the authors do not cite Doron Aurbach's seminal work on the use of AFM to study SEI.

Response: We thank the reviewer for the comments. Relevant AFM studies of Doron Aurbach²²⁻²⁵ on the SEI film at the electrode/electrolyte interface are of extensive and far-reaching significance in the researches on the interphasial evolution, property and corresponding mechanism, which have been cited and updated as ref. 24, 25, 31 and 32 in references and accordingly highlighted in the corresponding main text of the manuscript in page 3 line 28.

8. Some improvements in the english language are needed. While the authors' meaning can be understood in most places, there are also a number of places where more clarity is needed.

Response: We are sorry that a number of interpretations in the manuscript are unclear and confused. We have carefully examined throughout the whole manuscript, and the presentation of English language and the ambiguous statements have been accordingly improved and updated.

Reviewer #2 (Remarks to the Author):

This very interesting manuscript describes SEI formation and growth for lithiation and delithiation of a monolayer of MoS₂ in presence of two types of liquid electrolytes. The nature of the ML MS₂ makes the system ideal for this type of study and certainly the various stages are very well analyzed and discussed. The results are extremely useful for understanding of the SEI not only on this material but also in others. I suggest that before acceptance, the authors comment on the following aspects of their work.

1. The authors claim that the ultra-thin FEC film helps controlling the reactivity and therefore the number of reduction reactions in successive cycles tends to decrease. During delithiation the discussion after Figure 2 indicates that both film thickness and height of the NPs are reduced. Are these SEIs partially dissolved in the electrolyte? Or is it a reorganization of the structure of both the LiF film and the NPs? Please explain and show any evidence of either phenomena.

Response: We thank the reviewer for the comments and suggestions. Among the *in situ* AFM studies, we have visually found that both thickness of the FEC-derived SEI film and height of the NP-shaped one are reduced after delithiation. In order to quantitatively investigate the thickness change of two distinguishable SEI films during the lithiation/delithiation processes, we have selected 20 successive AFM images in one experiment for further quantified measurements (the first 10 images are captured upon lithiation, and the residual 10 images belong to delithiation process, $\Delta t = 3$ min). Specific statistics are shown in Supplementary Table S3, and the corresponding graph is shown in Supplementary Figure S7. From the statistical results, we believe that the thickness of the two SEI films both decrease after delithiation, whereas accompanying with different dynamic behaviors. For NP-shaped SEI film, it exhibits a faster kinetics at the initial growth with the nanoparticle size rapidly growing from 7.8 ± 1.6 nm to 10.5 ± 1.7 nm, and then remains substantially with the size of 11.8 ± 2.1 nm in the subsequent lithiation. Nevertheless, the ultra-thin FEC-derived SEI film almost maintains a thickness of 1.8 ± 0.9 nm upon lithiation, revealing the interphasial homogeneity of the FEC-derived SEI film. During the delithiation process, the thickness of the NP-shaped SEI film instantly decreases from 11.6 ± 1.7 nm to 10.1 ± 1.8 nm, and then remains stable at 9.2 ± 1.2 nm. However, the FEC-derived SEI film substantially maintains with the thickness of 1.7 ± 0.2 nm. Such different dynamic behaviors of NP-shaped and FEC-derived SEI films further distinctly elucidate the differences in essence and properties of two SEI films upon cycles. The relevant *in situ* AFM investigation and statistics manifest that the two distinguishable SEI films are independent of each other, so we could attribute the reduction in thickness after delithiation to the partial dissolution of these SEIs in the electrolyte.

Accordingly, we have added Supplementary Figure S7 and Table S3 in Supplementary Information, and the corresponding explanation has also been provided in the highlighted main text of manuscript in page 10 line 5, as “Discrepant dynamic behaviors of the NP-shaped one and FEC-derived one upon cycles further distinctly elucidate the differences

in essence and properties of the two distinguishable SEI films (thickness-cycle number graph and corresponding statistics are respectively shown in Supplementary Fig. S7 and Table S3).”.

Supplementary Figure S7. Quantitative measurements of the thickness of NP-shaped and FEC-derived SEI films upon lithiation/delithiation processes.

Time (min)	Thickness of FEC-derived SEI film (nm)			Average (nm)	Stdevp (nm)	Thickness of NP-shaped SEI film (nm)			Average (nm)	Stdevp (nm)
	1	2	3			1	2	3		
3	0.87	0.9	0.98	0.92	0.04	9.11	5.27	9.12	7.83	1.57
6	1.02	1.05	1.18	1.08	0.06	10.11	8.43	13.08	10.54	1.66
9	1.09	1.11	1.4	1.20	0.12	10.62	8.32	14.22	11.05	2.10
12	1.28	1.35	1.56	1.40	0.10	10.68	9.42	14.98	11.69	2.06
15	1.25	1.56	1.99	1.60	0.26	10.46	10.06	14.78	11.77	1.85
18	1.21	1.66	2.18	1.68	0.34	9.95	10.26	14.34	11.52	1.73
21	1.45	1.73	2.18	1.79	0.26	10.14	10.54	14.13	11.60	1.55
24	1.46	1.99	2.02	1.82	0.22	9.61	10.8	14.11	11.51	1.65
27	1.53	1.76	2.21	1.83	0.24	9.64	10.81	13.59	11.35	1.43
30	1.68	2.09	2.54	2.10	0.30	9.6	10.96	14.31	11.62	1.71
33	1.47	1.87	2.5	1.95	0.37	7.43	10.39	12.6	10.14	1.83
36	1.42	1.67	2.06	1.72	0.23	7.95	10.57	12.37	10.30	1.57
39	1.66	1.8	2.06	1.84	0.14	7.45	10.48	11.9	9.94	1.61
42	1.46	1.79	2.1	1.78	0.23	7.09	10.61	12	9.90	1.79
45	1.36	1.78	2.01	1.72	0.23	7.66	9.76	10.97	9.46	1.18
48	1.44	1.69	1.97	1.70	0.19	7.29	9.81	11.45	9.52	1.48

51	1.43	1.64	1.87	1.65	0.16	7.03	9.88	10.93	9.28	1.43
54	1.38	1.69	1.97	1.68	0.21	7.23	9.4	10.45	9.03	1.16
57	1.42	1.65	1.89	1.65	0.17	7.21	9.66	10.03	8.97	1.08
60	1.47	1.59	1.75	1.60	0.10	7.34	9.43	10.71	9.16	1.20

Supplementary Table S3. Correspondingly specific statistics of the thickness measurements of two distinguishable NP-shaped and FEC-derived SEI films.

2. How does the thin film thickness (LiF) grow in successive cycling?

Response: We thank the reviewer for the comments. In order to monitor the thickness variation of the ultra-thin FEC-derived SEI film upon cycles, we additionally performed an *in situ* AFM test during 10 cycles. Images after lithiation/delithiation in each cycle (Supplementary Figure S5) has been quantificationally measured and the graph of thickness-cycle number (Supplementary Figure S6) has been accordingly obtained, and the associated statistics have also been shown in Supplementary Table S2. The experimental results manifest that during the cycles, the thickness of the FEC-derived SEI film increase to a certain extent from 1.9 ± 0.3 nm of the 1st cycle to 2.6 ± 0.2 nm of the 10th cycle, but the change in overall is not apparent, revealing the ultra-thin nature and interphasial homogeneity of the FEC-derived SEI film.

Supplementary Figure S5, S6 and Table S2 have been additionally appended in Supplementary Information, and the relevant explanation has also been provided in the highlights of page 9 line 16 of manuscript, as “During *in situ* AFM cycles, the thicknesses of FEC-derived SEI film ranging from 1.4 nm to 3.1 nm with an average value of 2.4 ± 0.2 nm behave no conspicuous variation (detailed AFM images, thickness-cycle number graph and corresponding statistics are respectively shown in Supplementary Fig. S5, S6 and Table S2). Adequately revealing the ultra-thin nature and interphasial homogeneity of the FEC-derived SEI film upon cycles.”.

Supplementary Figure S5. *In situ* AFM images of the monolayer MoS₂ electrode/electrolyte interface after lithiation/delithiation in each cycle in the electrolyte with 10 wt% FEC upon 10 cycles. The scale bar is 300 nm.

Supplementary Figure S6. Quantitative measurements of the thickness of FEC-derived SEI film after lithiation/delithiation in each cycle upon 10 cycles.

Lithiation			
Cycle number	Thickness of FEC-derived SEI (nm)	Average	Stdevp
1	2.05	2.05	0.15
2	2.10	2.10	0.15
3	2.70	2.70	0.15
4	2.65	2.65	0.15
5	2.25	2.25	0.15
6	2.55	2.55	0.15
7	2.75	2.75	0.15
8	2.90	2.90	0.15
9	2.55	2.55	0.15
10	2.60	2.60	0.15

	1	2	3	(nm)	(nm)
1	2.29	1.94	1.96	2.06	0.14
2	2.37	2.09	1.88	2.11	0.17
3	2.72	2.68	2.7	2.70	0.01
4	2.78	2.73	2.53	2.68	0.09
5	2.52	2.14	2.15	2.27	0.15
6	2.92	2.42	2.46	2.60	0.20
7	2.96	2.68	2.58	2.74	0.14
8	3.17	2.97	2.52	2.89	0.24
9	2.94	2.57	2.26	2.59	0.24
10	3	2.39	2.44	2.61	0.24

Delithiation

Cycle number	Thickness of FEC-derived SEI (nm)			Average (nm)	Stdevp (nm)
	1	2	3		
1	2.02	1.57	1.4	1.66	0.23
2	2.48	2.18	2.16	2.27	0.13
3	2.46	2.08	1.92	2.15	0.20
4	2.54	1.91	2.02	2.16	0.24
5	2.56	2.07	1.74	2.12	0.29
6	2.63	2.12	2.25	2.33	0.19
7	2.97	2.6	2.18	2.58	0.28
8	2.74	2.59	2.14	2.49	0.22
9	2.43	2.35	2.11	2.30	0.12
10	2.9	2.5	2.3	2.57	0.22

Supplementary Table S2. Correspondingly specific statistics of the thickness measurements of the FEC-derived SEI film after lithiation/delithiation of each cycle in 10 cycling processes.

3. How does the phase transition affect the SEI formation if at all? Does it affect lithiation or delithiation? How?

Response: We thank the reviewer for the comments. Based on the electrochemical tests and *in situ* AFM experiments, we can conclude that the phase transition process, which is the origin of wrinkle-structure networks, occurs at the lithiation potential (~ 1.1 V), whereas the SEI film-forming potentials are more positive than lithiation (~ 1.77 V of FEC-derived SEI and

~1.39 V of NP-shaped SEI). So we can predicatively believe that the phase transition has no effect on the SEI formation in the first cycle.

Furthermore, regarding the driving force of the phase transition and wrinkling morphology, it has been reported that, at the initial lithiation of MoS₂ electrode, a large amount of Li ions embed into the tetrahedral coordination center of the S-S, resulting in the charge transfer from intercalated Li⁺. Due to the co-intercalation of ions and electrons, the coordination structure of metal Mo is converted from the trigonal prism to octahedron, and the corresponding space group is transformed from *P6₃/mmc* to *P-3m1*,¹⁵ thus producing 2H-1T phase transition in order to reduce the energy of the system⁹. Intrinsic semiconductivity of 2H MoS₂ is accordingly changed into metallic 1T Li_xMoS₂, and therefore, a charge density wave (CDW) generates in the 2D plane caused by the appearance of superlattices due to the instability of the low-dimensional metal¹⁶. Associated first-principles calculations have elucidated that the essence of the phase transition upon lithiation is the interaction of intrinsic doping and electron-phonon coupling⁸. HRTEM¹⁶ and STEM¹⁷ have also been performed to achieve the *in situ* monitoring of the in-plane relative gliding and dynamic phase boundary movement during ion implantation. And the relevant shear mechanism¹⁶ and compressive stress⁴ in 2D planes of MoS₂ have also been proposed in phase transition process, that is, the direct causes of wrinkles morphology upon lithiation. So far, the origin and source of phase transition and planar nanofolds morphology caused by lithiation could be clearly explained. Such phase transition process will cause the change from semiconducting to metallic of the MoS₂ electrode, which can increase the conductivity of the electrodes and thus may partially affect the lithiation/delithiation processes upon cycles.

The according interpretation and discussion has been added in highlights of the page 7 line 5 of the revised manuscript, as “Regarding the driving force of the wrinkles, it has been reported that the in-plane relative gliding and dynamic phase boundary movement induced by the charge transfer from intercalated Li⁺, which is intrinsically caused by the interaction of electron doping and electron-phonon coupling, finally result in the coordination structure of metal Mo converting from the trigonal prism to octahedron, thus generate the well-known 2H-1T phase transition to reduce the system energy. Changes in the lattice parameters and structures further produce biaxial compressive stresses and strains in the plane after being constrained by the hard substrate, leading to the distortion by microstrains, and subsequently accompany with the planar nanofolds and ultimately develop to a dense wrinkle-structure network.”.

4. Do the wrinkles grow upon cycling? How many cycles could this type of electrode be used for each electrolyte?

Response: We thank the reviewer for the comments. In order to investigate the changes in height of wrinkles upon cycles, we have performed 10 cycles of *in situ* AFM monitoring in both electrolytes with/without 10 wt% FEC. Quantitative measurements have been performed on the images after lithiation/delithiation processes of each cycle (Supplementary Figure S5 shows the AFM images of FEC-containing system, and Supplementary Figure S8 exhibits that of FEC-free system), and the corresponding graph of wrinkle heights-cycle number has been acquired (Supplementary Figure S9), and the relevant statistics have also

been shown in Supplementary Table S4. The experimental results indicate that, for FEC-free system, the average height of wrinkles increases from 5.1 ± 0.4 nm of the 1st cycle to 9.0 ± 1.7 nm of the 10th cycle, appearing as a continuously growing and subsequently maintaining trend throughout the cycles. Nevertheless, in FEC-containing system, there is no significant increase in the wrinkle heights with an overall retention of 4.1 ± 0.6 nm, elucidating the surface effect of FEC additive and the cycle stability of electrodes. The larger decay of the wrinkle heights after delithiation in FEC-containing system, further indicating a better reversibility upon cycles.

Supplementary Figure S8, S9 and Table S4 have been additionally appended in Supplementary Information, and the relevant explanation has also been provided in the highlights of page 11 line 6 of manuscript, as “Quantitatively measured heights of wrinkles in two systems upon cycles further demonstrate a sparser and more stable wrinkling network in the electrolyte with 10 wt% FEC additive (detailed AFM images of FEC-containing/free systems are respectively shown in Supplementary Fig. S5 and S8, thickness-cycle number graph and corresponding statistics are indicated in Supplementary Fig. S9 and Table S4).”.

In the present work, the monolayer MoS_2 was established as a model electrode to simultaneously meet the requirements of the *in situ* AFM observation and electrochemical conductivity. Such monolayer MoS_2 model electrodes are not suitable for the tests of electrochemical performance in conventional coin cells, so we mixed the MoS_2 powders with super P (conductive additive) and PVDF (binder material) in a ratio of 8:1:1 and then assembled into coin cells with the electrolytes of $[\text{BMP}]^+[\text{FSI}]^-$ containing 0.5 M LiFSI with/without 10wt % FEC for electrochemical testing. The corresponding results of cycling performance (as shown in Figure 2k of the manuscript) indicate that the discharge specific capacity of the FEC-free system decreases from 1024 mAh/g of 1st cycle to 39.2 mAh/g of 100th cycle, whereas that of FEC-containing system declines from 1015 to 164.6 mAh/g, revealing a better cycling capacity and stability of batteries in the electrolyte with FEC additive.

Supplementary Figure S8. *In situ* AFM images of the monolayer MoS₂ electrode/electrolyte interface after lithiation/delithiation in each cycle in the electrolyte without FEC upon 10 cycles. The scale bar is 500 nm.

Supplementary Figure S9. Quantitative measurements of the height of wrinkles after lithiation/delithiation in each cycle in the electrolyte with/without 10 wt% FEC upon 10 cycles.

Lithiation						
Cycle	Height of wrinkles in	Average	Stdevp	Height of wrinkles	Average	Stdevp

number	FEC-free system (nm)			(nm)	(nm)	in FEC-containing system (nm)			(nm)	(nm)
	1	2	3			1	2	3		
1	5.19	5.13	5.05	5.12	0.05	4.66	3.48	4.29	4.14	0.43
2	7.62	6.14	6.39	6.72	0.56	4.55	3.61	6.59	4.92	1.08
3	7.25	5.77	7.59	6.87	0.68	5.23	3.87	6.52	5.21	0.94
4	7.73	6.08	7.26	7.02	0.60	4.55	3.5	4.18	4.08	0.38
5	8.22	7.3	8.75	8.09	0.52	5.13	3.7	4.63	4.49	0.51
6	8.83	6.58	8.32	7.91	0.83	5.03	3.73	3.32	4.03	0.63
7	8.55	5.54	8.9	7.66	1.31	4.25	3.06	3.21	3.51	0.46
8	10.11	7.23	9.82	9.05	1.12	5.2	3.41	3.86	4.16	0.66
9	10.27	8.51	9.96	9.58	0.66	5.22	3.63	4.06	4.30	0.58
10	10.29	7.35	10.84	9.49	1.33	4.74	3.53	4.8	4.36	0.51

Delithiation

Cycle number	Height of wrinkles in FEC-free system (nm)			Average (nm)	Stdevp (nm)	Height of wrinkles in FEC-containing system (nm)			Average (nm)	Stdevp (nm)
	1	2	3			1	2	3		
1	5.37	5.25	4.41	5.01	0.37	4.05	3.04	3.24	3.44	0.38
2	7.71	5.48	6.13	6.44	0.81	4.61	3.69	3.38	3.89	0.45
3	7.38	5.49	7.07	6.65	0.72	5.35	3.92	4.36	4.54	0.52
4	7.45	5.57	7.28	6.77	0.74	4.34	3.67	3.04	3.68	0.46
5	8.2	6.51	8.5	7.74	0.76	4.72	3.3	3.5	3.84	0.54
6	7.88	6.67	8.83	7.79	0.77	4.52	3.11	2.85	3.49	0.64
7	8.3	6.41	8.7	7.80	0.86	4.22	2.51	3.56	3.43	0.61
8	10.15	7.48	9.15	8.93	0.95	4.33	3.03	4.31	3.89	0.53
9	9.15	7.22	9.22	8.53	0.80	4.16	3.74	4.04	3.98	0.15
10	10.56	5.84	8.94	8.45	1.70	4.08	3.2	4.51	3.93	0.47

Supplementary Table S4. Correspondingly specific statistics of the height measurements of the wrinkles after lithiation/delithiation of each cycle in FEC-free and FEC-containing systems upon 10 cycling tests.

References

- 1 Lee, Y. H. *et al.* Synthesis of large-area MoS₂ atomic layers with chemical vapor deposition. *Adv. Mater.* **24**, 2320-2325 (2012).
- 2 Chen, W. *et al.* Oxygen-assisted chemical vapor deposition growth of large single-crystal and high-quality monolayer MoS₂. *J. Am. Chem. Soc.* **137**, 15632-15635 (2015).
- 3 Ling, X. *et al.* Role of the seeding promoter in MoS₂ growth by chemical vapor deposition. *Nano Lett.* **14**, 464-472 (2014).
- 4 Peng, J. *et al.* Very large-sized transition metal dichalcogenides monolayers from fast exfoliation by manual shaking. *J. Am. Chem. Soc.* **139**, 9019-9025 (2017).
- 5 Azhagurajan, M., Kajita, T., Itoh, T., Kim, Y. G. & Itaya, K. In situ visualization of lithium ion intercalation into MoS₂ single crystals using differential optical microscopy with atomic layer resolution. *J. Am. Chem. Soc.* **138**, 3355-3361 (2016).
- 6 Lacey, S. D. *et al.* Atomic force microscopy studies on molybdenum disulfide flakes as sodium-ion anodes. *Nano Lett.* **15**, 1018-1024 (2015).
- 7 Spiecker, E. *et al.* Self-assembled nanofold network formation on layered crystal surfaces during metal intercalation. *Phys. Rev. Lett.* **96**, 086401 (2006).
- 8 Schwingenschlogl, U. Origin of the phase transition in lithiated molybdenum disulfide. *ACS nano* **8**, 11447-11453 (2014).
- 9 Leng, K. *et al.* Phase restructuring in transition metal dichalcogenides for highly stable energy storage. *ACS Nano* **10**, 9208-9215 (2016).
- 10 Chhowalla, M. *et al.* The chemistry of two-dimensional layered transition metal dichalcogenide nanosheets. *Nat. Chem.* **5**, 263-275 (2013).
- 11 Py, M. A., and Haering, R. R. Structural destabilization induced by lithium intercalation in MoS₂ and related compounds. *Can. J. Phys.* **61**, 76-84, (1983).
- 12 Mattheiss, L. F. Band structures of transition-metal-dichalcogenide layer compounds. *Phys. Rev. B* **8**, 3719-3740 (1973).
- 13 Wang, J., Wei, Y., Li, H., Huang, X. & Zhang, H. Crystal phase control in two-dimensional materials. *Sci. Chi. Chem.* **61**, 1227-1242 (2018).
- 14 Duerloo, K. A., Li, Y. & Reed, E. J. Structural phase transitions in two-dimensional Mo- and W-dichalcogenide monolayers. *Nat. Commun.* **5**, 4214 (2014).
- 15 Huang, Q., Wang, L., Xu, Z., Wang, W. & Bai, X. In-situ TEM investigation of MoS₂ upon alkali metal intercalation. *Sci. Chi. Chem.* **61**, 222-227 (2017).
- 16 Wang, L., Xu, Z., Wang, W. & Bai, X. Atomic mechanism of dynamic electrochemical lithiation processes of MoS₂ nanosheets. *J. Am. Chem. Soc.* **136**, 6693-6697 (2014).
- 17 Lin, Y. C., Dumcenco, D. O., Huang, Y. S. & Suenaga, K. Atomic mechanism of the semiconducting-to-metallic phase transition in single-layered MoS₂. *Nat. Nanotechnol.* **9**, 391-396 (2014).
- 18 Tokranov, A., Sheldon, B. W., Lu, P., Xiao, X. & Mukhopadhyay, A. The origin of stress in the solid electrolyte interphase on carbon electrodes for Li ion batteries. *J. Electrochem. Soc.* **161**, A58-A65 (2013).

- 19 Winter, M. The solid electrolyte interphase-the most important and the least understood solid electrolyte in rechargeable Li batteries. *Z. Phys. Chem.* **223**, 1395-1406 (2009).
- 20 Xu, K. Electrolytes and interphases in Li-ion batteries and beyond. *Chem. Rev.* **114**, 11503-11618 (2014).
- 21 Gu, Y. *et al.* Designable ultra-smooth ultra-thin solid-electrolyte interphases of three alkali metal anodes. *Nat. Commun.* **9**, 1339 (2018).
- 22 Koltypin, M., Cohen, Y. S., Markovsky, B., Cohen, Y., & Aurbach, D. The study of lithium insertion-deinsertion processes into composite graphite electrodes by in situ atomic force microscopy (afm). *Electrochem. Commun.* **4**, 17-23 (2002).
- 23 Aurbach, D., Maxim Koltypin, A., & Teller, H. In situ AFM imaging of surface phenomena on composite graphite electrodes during lithium insertion. *Langmuir* **18**, 27-43 (2002).
- 24 Cohen, Y. S., Cohen, Y., & Aurbach, D. Micromorphological studies of lithium electrodes in alkyl carbonate solutions using in situ atomic force microscopy. *J. Phys. Chem. B* **104**, 12282-12291 (2000).
- 25 Aurbach, D., & Cohen, Y. The application of atomic force microscopy for the study of Li deposition processes. *J. Electrochem. Soc.* **143**, 3525-3532 (1996).

Reviewers' comments:

Reviewer #1 (Remarks to the Author):

The authors have provided reasonable responses to most of the points that I raised. However, I have the following additional requests:

1. The additional explanation of stress-driven wrinkling is helpful. However, this is still rather speculative and it should be presented as such. Also, one important issue here that should be clarified is that the proposed wrinkling requires either deformation of the underlying material, or debonding from the underlying material. The possibility of deformation can be readily checked with basic wrinkling mechanics criteria (this requires values for the in-plane strain, the film thickness, and the modulus of the materials .. it appears that estimates for all of these are available).
2. A great deal of the explanations that are provided to the reviewers (in response to comments) have not been used to modify the paper. Most of these explanations should be added in some form to the paper to provide appropriate clarification.
3. A lot of the new text that is included (highlighted in yellow) is not well written. For example, there are some extremely long and confusing sentences here. Please review this and improve the english where necessary.

Reviewer #2 (Remarks to the Author):

The authors have responded satisfactorily to the comments of the Reviewers. I recommend acceptance as is.

Reviewers' comments:

Reviewer #1 (Remarks to the Author):

The authors have provided reasonable responses to most of the points that I raised. However, I have the following additional requests:

We appreciate the reviewer for all the suggestions and comments which are greatly helpful in improving our manuscript.

1. The additional explanation of stress-driven wrinkling is helpful. However, this is still rather speculative and it should be presented as such. Also, one important issue here that should be clarified is that the proposed wrinkling requires either deformation of the underlying material, or debonding from the underlying material. The possibility of deformation can be readily checked with basic wrinkling mechanics criteria (this requires values for the in-plane strain, the film thickness, and the modulus of the materials.. It appears that estimates for all of these are available).

Response: We thank the reviewer for the comment and suggestion. We have further improved the presented explanation and provided additional experimental data in the revised manuscript. We hope the modified discussions have become more clarified in the current version.

Regarding the issue whether the proposed wrinkling requires deformation of the underlying material, the explanations are provided as follows: 1. Si (100) electrode, as the underlying material of MoS₂ in this work, has been taken as working electrode for *in situ* AFM measurement in the same electrolyte. It is clear that no deformation of Si (100) electrode occurs during the potential range of wrinkle formation in this work, from OCP (2.64 V) to cathodic 1 V, as shown in the additionally added Supplementary Figure S5. It's accordingly indicated that the Si electrode itself is unable to generate deformation to induce the wrinkling on MoS₂ layer in the presented potential range; 2. The related DMT Modulus image of wrinkle-structure networks on monolayer MoS₂ is additionally shown in Supplementary Figure S6, manifesting that the underlying Si substrate with a higher modulus has almost no distinct deformation upon lithiation; 3. As shown in the Figure 1 and Figure 3, the wrinkling morphology clearly appear in both monolayer and multilayer MoS₂ systems, further suggesting that the wrinkle formation is mostly correlated to the intrinsic properties of MoS₂ rather than the deformation of underlying material.

In addition, regarding the issue whether the proposed wrinkling requires debonding from the underlying material, the driving force of the generation of debonding could be taken into account for understanding the correlations. It's known that intercalation reaction is one of the methods for achieving the debonding from layered 2D materials. In other word, in the present work, debonding from the underlying material could be induced by extensive intercalation of Li⁺ ions into the intermediate layer between monolayer MoS₂ and underlying Si substrate upon lithiation. Understanding the pathway of the Li⁺ migration can further help us to estimate this issue. Clearly, when it comes to the migration path of Li⁺ in the case of monolayer MoS₂ on Si substrate, it can be generally proposed as two ways. One is that Li⁺ ions directly adsorb on the surface of MoS₂, and the other one is that interpose between the

MoS₂ and substrate from the edge sites of the monolayer MoS₂. Firstly, if a majority of Li⁺ ions directly adsorb on the electrode, an even germination of wrinkles will take place. For another, if Li⁺ intercalation mostly happens between the MoS₂ and substrate, the growth of wrinkle-structure networks will obviously exhibit as the edge-to-center mode. The obtained *in situ* AFM results demonstrate that the wrinkle-structure uniformly generate on the electrode surface upon lithiation, instead of an edge-to-center growth mode. Accordingly, it could be considered that the majority of Li⁺ ions prefer to directly adsorb on the monolayer MoS₂ surface for further reaction with MoS₂ rather than intercalate into the intermediate layer at the given potential range. Consequently, the debonding from the underlying substrate could not take place due to weak intercalation of Li⁺ ions. In other words, the observed wrinkling may not require the debonding from the underlying substrate due to no apparent debonding occurring, but the released stress upon wrinkle formation may partially weaken the interaction force between the MoS₂ electrode and Si substrate. Therefore, based on the above analysis, it's drawn that the possibility of deformation of the underlying material or debonding from the underlying material in inducing wrinkle formation could be ruled out in the present system.

Accordingly, involved AFM images have been additionally introduced in Supplementary Figure S5 and S6, and corresponding explanation about the issue whether the proposed wrinkling requires deformation of the underlying material or debonding from the underlying material, has been added in yellow highlighting from the page 7 line 29 to page 8 line 24 in revised manuscript.

Supplementary Figure S5. *In situ* AFM images of Si (100) electrode in [BMP]⁺[FSI]⁻ containing 0.5 M LiFSI electrolyte (a) at OCP, and (b) charged from 1.23 V to 1.03 V. The scale bar is 200 nm.

Supplementary Figure 6. Morphology and DMT Modulus of the lithiated monolayer MoS₂ with underlying Si. The AFM images of (a) topography, (b) DMT Modulus and (c) deformation of the wrinkle-like MoS₂ and Si substrate at cathodic 1.0 V. The scale bar is 400 nm.

2. A great deal of the explanations that are provided to the reviewers (in response to comments) have not been used to modify the paper. Most of these explanations should be added in some form to the paper to provide appropriate clarification.

Response: We thank the reviewer for the comment and suggestion. We have additionally added the explanations in the revised manuscript based on the response to comments, as highlighted in yellow in page 6-10 of the revised manuscript.

3. A lot of the new text that is included (highlighted in yellow) is not well written. For example, there are some extremely long and confusing sentences here. Please review this and improve the english where necessary.

Response: We thank the reviewer for the comment and suggestion. We have carefully corrected the confusing sentences especially in the new text with yellow highlighting and polished the language throughout the whole revised manuscript.

REVIEWERS' COMMENTS:

Reviewer #1 (Remarks to the Author):

In response to my previous review, the authors have provided appropriate improvements.